# Fibrin Clot Formation and Lysis in Plasma

**DOI:** 10.3390/mps3040067

**Published:** 2020-09-25

**Authors:** Julie Brogaard Larsen, Anne-Mette Hvas

**Affiliations:** 1Thrombosis and Haemostasis Research Unit, Department of Clinical Biochemistry, Aarhus University Hospital, Palle Juul-Jensens Boulevard 99, 8200 Aarhus, Denmark; am.hvas@dadlnet.dk; 2Department of Clinical Medicine, Health, Aarhus University, Palle Juul-Jensens Boulevard 82, 8200 Aarhus, Denmark

**Keywords:** blood coagulation, fibrinolysis, clot formation and lysis assay

## Abstract

Disturbance in the balance between fibrin formation and fibrinolysis can lead to either bleeding or thrombosis; however, our current routine coagulation assays are not sensitive to altered fibrinolysis. The clot formation and lysis assay is a dynamic plasma-based analysis that assesses the patient’s capacity for fibrin formation and fibrinolysis by adding an activator of coagulation as well as fibrinolysis to plasma and measuring ex vivo fibrin clot formation and breakdown over time. This assay provides detailed information on the fibrinolytic activity but is currently used for research only, as the assay is prone to inter-laboratory variation and as it demands experienced laboratory technicians as well as specialized personnel to validate and interpret the results. Here, we describe a protocol for the clot formation and lysis assay used at our research laboratory.

## 1. Introduction

Fibrinolysis is the ongoing physiological process of fibrin clot breakdown and is normally tightly regulated to keep the balance between fibrin formation and breakdown (Figure 1) [1]. This balance secures hemostasis in the case of vessel wall damage while preventing excess fibrin formation and obstruction of blood supply to end organs. Upon activation of the coagulation system, circulating fibrinogen is converted to fibrin, and the fibrin clot is further stabilized via fibrin cross-linking by coagulation factor (F) XIIIa. Plasmin is the main fibrinolytic protease and circulates in the blood in its zymogen form plasminogen. The cross-linked fibrin provides a binding surface for plasminogen, which is then converted into plasmin by tissue-type or urokinase-type plasminogen activator (tPA/uPA), of which tPA is the most abundant. Fibrinolysis is regulated by the anti-fibrinolytic proteins α2-antiplasmin, plasminogen activator inhibitor (PAI)-1 and -2, and thrombin-activatable fibrinolysis inhibitor (TAFI) [2,3,4]. Finally, the structure of the fibrin clot itself influences fibrinolysis, as denser fibrin clots with smaller pores have been found less susceptible to lysis, probably because binding of plasminogen and tPA to fibrin is impeded by smaller pore size [5,6].

Altered fibrinolysis occurs in a range of clinical settings. Hyperfibrinolysis can lead to severely increased bleeding tendency [7], while hypofibrinolysis is associated with an increased thrombosis risk [8]. However, current routine coagulation assays, such as the activated partial thromboplastin time (aPTT) and prothrombin time (PT), are not sensitive to fibrinolysis. High circulating fibrin degradation products indicate increased fibrin turnover but will usually reflect increased procoagulant activity and fibrin formation more than hyperfibrinolysis. Thus, more sensitive and specific biomarkers to assess fibrinolytic capacity are necessary for research and the clinical laboratory.

The plasma-based clot formation and lysis assay allow for a detailed assessment of fibrin formation and breakdown capacity. Several different versions of the assay have been published [9,10,11,12,13,14,15]. The common principle is that citrated, platelet-poor plasma (PPP) is mixed with an activator of coagulation, usually recombinant tissue factor (TF) or thrombin, as well as phospholipids and calcium to induce fibrin formation. Simultaneously, tPA or another plasminogen activator is added to induce clot lysis. The assay employs a turbidimetric principle, as the fibrin network is first formed and then lysed in the well, turbidity increases and subsequently decreases. Absorbance is registered continuously over a specified time period (e.g., 1.5 h), resulting in the formation of the clot-lysis curve (Figure 2), from which the following parameters can be derived: time to initial fibrin formation (lag phase), maximum absorbance (peak fibrin concentration in well), integral or area under curve (net fibrin formation), and time from peak to 50% lysis of the clot (50% lysis time).

The clot-lysis curve shape and reference values for derived parameters vary considerably with the type and final concentrations of coagulation activators, tPA and Ca^2+^ (see Section 4). The present protocol describes the experimental design and reference values used at the Thrombosis and Haemostasis Research Unit, Aarhus University Hospital, Aarhus, Denmark.

## 2. Materials

### 2.1. Patient Preparation

Blood sampling and preparation of PPP, see Section 6.1, Section 6.2 and Section 6.3.


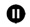
**PAUSE STEP** After preparation, PPP can be stored at −80 °C for 12 months.

### 2.2. Reagents

Human recombinant TF (Dade^®^ Innovin^®^, Siemens Healthcare, Diagnostics Aps, Ballerup, Denmark. Ref.no.: B4212-40)Human recombinant tPA, lyophilized, 100 µg (Calbiochem^®^, Sigma-Aldrich, Merck, Darmstadt, Germany. Cat. no.: 612200)Phospholipids 500 µM (Rossix, Mölndal, Sweden. Ref.no.: PL604T)HEPES buffer, 20 mM, NaCl 150 mM, pH 7.4 (Ampliqon, Odense, Denmark)HEPES, 20 mM, NaCl 150 mM, CaCl_2_ 200 mM, pH 7.4 (Ampliqon, Odense, Denmark)Bovine serum albumin (BSA), lyophilized (>98%) (Sigma-Aldrich, Merck, Darmstadt, Germany. Cat. no.: A70-30-100a)Controls: Pooled normal plasma (PrecisionBiologic, Cryocheck^TM^, Haemochrom Diagnostica, Frederiksberg, Denmark. Cat.no.: CCN-10)Demineralized water (resistivity 18.2 MΩ × cm at 25 °C) to dissolve TF and tPA and flush dispensers

Solutions to be added in the well:10 µL HEPES buffer (see Section 3.1)10 µL phospholipid 60 µM solution (see Section 6.5), Target final concentration in well = 4 µM20 µL TF B 1:665 solution (see Section 6.5), Target final dilution in well = 1:500070 µL PPP (see Section 2.1)20 µL tPA 870 ng/mL solution (see Section 6.5), Target final concentration in well = 116 ng/mL20 µL HEPES-Ca (see Section 3.1), Target Ca^2+^ concentration in well = 26.7 mMTotal volume in well = 150 µL

### 2.3. Equipment

96-well plate (Nunc ImmunoPlate, Thermo Fisher Scientific, Roskilde, Denmark. Cat. no.: 442404)5 mL and 10 mL tubes for preparation of reagentsVictor Reader X4 (Perkin Elmer, Waltham, MA, USA)Two automatic 1-channel dispensers (PerkinElmer, Waltham, MA, USA)Software: 2030 WorkOut and WorkOut 2.5 (Perkin Elmer, Waltham, MA, USA)

OPTIONAL: Automatic dispensers and software. See Section 6.6.

## 3. Procedure

### 3.1. Reconstitution of Reagents and Preparation of Buffers

Reconstitute TF in 4 mL distilled water, aliquot, and store at −80 °C until use.Reconstitute tPA in 1 mL distilled water. Mix gently, aliquot, and store at −80 °C until use.Prepare 1% *w*/*v* BSA in HEPES, aliquot in 4 mL portions, and store at −20 °C until use.


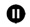
**PAUSE STEP** Reconstituted TF, tPA, and HEPES with BSA can be stored for several months. See Section 6.4.

### 3.2. Preparation of Victor Protocol

Setup of protocol in WorkOut, see Section 6.7.

### 3.3. Preparation for Analysis (30 min)

Turn on Victor reader and computer. Open software “PerkinElmer”. Start heating to 37 °C. Start WorkOut; choose the appropriate protocol name.Make a note of plate layout with ID numbers; plasma samples in **duplicate**. Controls should be positioned at B1, B2, G11, and G12.Collect plasma samples, controls, and TF in −80 °C freezer.Collect HEPES-BSA1% from freezer, thaw in a water bath at 37 °C for 1 min, and then thaw in a 5 °C cooler.Collect HEPES and HEPES-Ca from the cooler.


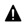
**CRITICAL STEP** Buffers should reach room temperature before reagent solutions are mixed with plasma.

6.Thaw plasma samples and controls in a water bath at 37 °C for 5 min.7.
**Prepare plasma samples and controls:**
Remove samples and controls from a water bath and refer to a dry tray.Mix samples and controls by gently inverting them five times.Spin samples and controls in a micro-centrifuge at 15,000 *g* for 3 min.
8.Label seven 5 mL tubes: “PL”, “TF A”, “TF”, “PLTFH”, “tPA A”, “tPA”, “HEPES-Ca” and one 10 mL tube: “HEPES”.9.Add relevant buffers to tubes:
“HEPES”: 7 mL HEPES“HEPES-Ca”: 4 mL HEPES-CaRemaining tubes: as detailed in Section 6.5.
10.**Prepare a 60 µM phospholipid solution** (tube: “PL”). See Section 6.5.11.**Prepare a 1:665 TF dilution** (tubes: “TF A” and “TF”). See Section 6.5.12.**Prepare a mix** of TF 1:665, phospholipids 60 µM, and HEPES (tube: PLTFH). See Section 6.5.13.**Prepare plate**:
Add 40 µL “PLTFH” to each well on plate.Add 70 µL of plasma sample or control to each well according to plate layout.Place plate in reader.

14.**Prepare tPA** 870 ng/mL solution (tubes: “tPA A”, “tPA”) as detailed in Section 6.5.


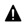
**CRITICAL STEP** Keep tPA at −80 °C until immediately before use, thaw at room temperature for 5 min. **Do not vortex**; shake gently.

15.
**Prepare dispensers:**
Flush dispensers with distilled water.Flush dispensers with air.Flush dispenser 1 with tPA 870 ng/mL solution.Flush dispenser 2 with HEPES-Ca.


### 3.4. Analysis (80 min)

On the computer screen, press “Start measurements”:
Dispenser 1 will now add 20 µL tPA to each well.Dispenser 2 will subsequently add 20 µL HEPES-Ca to each well. This activates coagulation.The plate will be shaken for 10 s.Reading will begin.
Reading (absorbance at 405 nm, 1 read/min for 80 min). See Section 6.6.


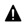
**CRITICAL STEP** After use, clean/flush both dispensers with a pipette and then flush both dispensers five times with distilled water to avoid clotting of the system.

### 3.5. Data Export

Create pictures of graphs for visual assessment and export raw data to Excel. See Section 6.8.

### 3.6. Results Validation

Inspect all graphs of duplicates visually. The same reproducible duplicate and almost full lysis should be present; if not, exclude samples from analysis (perform re-run). Check baseline correction and crossing point to assure that baseline and lag phase are calculated correctly. See Figure 2 and Figure 3 (top).In Excel: inspect all individual samples for peak and integral. We accept a coefficient of variation (CV) of 15% for all parameters.

Unexpected results and potential explanations:High CV% between duplicatesClotted sample. Inspect samples visually.Pipetting error.Flat curve (no derived parameters can be calculated), see Figure 3.Single sample, controls normal: Patient-related—see Table 1; clotted sample; pipetting error.Controls/entire plate: Faulty activation of coagulation: check TF reagent, HEPES-Ca.No or partial lysis only (integral and 50% lysis time cannot be calculated), see Figure 3.Single sample, controls normal: Patient-related—see Table 1.Controls/entire plate: Faulty activation of fibrinolysis: problems with tPA reagent; automatic pipetting error.

## 4. Expected Results and Interpretation

Table 1 shows definitions and interpretations of the parameters derived from the clot-lysis curve, with our local reference intervals.

The rate of fibrin formation and lysis, and hence the reference intervals for derived parameters, are very much dependent on the type and concentrations of activators used. Our group investigated clot-lysis using thrombin vs. TF as an activator in a cohort of 538 coronary artery disease patients [18]. Activation with TF resulted in higher net fibrin formation than thrombin, with higher maximum fibrin formation, higher integral, and longer lysis time (Table 2).

The assay is also sensitive to tPA concentration, as higher final tPA concentration increases net lysis, leading to decreased integral and shorter lysis times (Table 3).

This makes comparison difficult between laboratories. However, the variability of the assay also provides the opportunity to adjust the assay according to the specific research question or to investigate the effect of other factors on fibrinolysis by performing additional experiments.

It should be noted that considerable inter-laboratory variation has also been described even with the same protocol and reference plasma, which indicates that the clot-lysis assay is sensitive to even minor differences in equipment, reagents, and manual skills [19]. Therefore, thorough validation, including the establishment of local reference intervals, is necessary to implement the assay successfully in the research laboratory.

Altered fibrin formation and fibrinolytic capacity assessed by the clot-lysis assay has been described in a range of clinical conditions (Table 4). These findings highlight the contribution of the fibrinolytic system in the development of these conditions and for related adverse outcomes.

## 5. Summary and Conclusions

To summarize, we here provide a protocol for the fibrin clot formation and lysis assay as performed at the Thrombosis and Haemostasis Research Unit, Aarhus University Hospital, Denmark. Altered fibrinolysis may contribute to increased bleeding or thrombosis risk in a range of clinical conditions, and detailed assessment of plasmatic fibrinolytic capacity may support both research and clinical practice. Currently, inter-protocol and inter-laboratory variation, as well as differences in data analysis and reporting, challenge comparison between laboratories. Furthermore, the assay is labor-intensive, as only manual or semi-automated versions of the assay currently exist, and a high degree of skill is required to obtain acceptable precision and reproducibility. All these factors impede the implementation of the assay in clinical use. However, the clot formation and lysis assay provide a valuable research tool to characterize fibrinolytic capacity.

## 6. Notes

### 6.1. Patient Preparation

The presence of anticoagulant or antifibrinolytic drugs in the blood will influence the result.

### 6.2. Blood Sampling

Blood should be drawn from an antecubital vein using a 19 or 21 gauge needle with smooth venipuncture and minimal stasis in order to minimize endothelial and platelet activation and subsequent tPA and PAI-1 release. The use of a butterfly cannula is acceptable. Sodium citrate anticoagulated tubes (3.2%) should be used. Correct filling of the tube, to the mark pre-specified by the manufacturer is important to ensure the correct ratio of blood to anticoagulant. The first 1 mL should be discarded or used for other analyses to avoid spuriously high amounts of endothelial- or subendothelial-derived TF, tPA, and PAI-1 in the sample following the venipuncture and vessel wall. The remaining tubes should be gently inverted five times to ensure adequate mixing of blood and anticoagulant. If there are signs of clotting in the tube, it should be discarded. Visibly haemolysed samples are not suitable for analysis due to increased plasma calcium and adenosine diphosphate, which activates coagulation and platelets. Pronounced icterus and lipaemia may influence absorbance or turbidity; however, as baseline correction is performed automatically, samples with mild to moderate icterus or lipaemia are acceptable.

### 6.3. Preparation of PPP

Centrifugation should be performed at 3000× *g* for 25 min at room temperature within 1 hour after blood sampling. Plasma should be aliquoted into secondary tubes and frozen at −80 °C immediately after aliquoting and within 2 h of blood sampling at the latest. Avoid storage under cool conditions, as this may activate coagulation. After correct preparation, PPP should be stored at −80 °C before analysis. Repeated freeze-thawing affects the analysis in our experience and should be avoided. We have not tested the duration of stability at −80 °C, but based on the stability of other coagulation parameters, we expect acceptable stability for a minimum of one year.

### 6.4. Reagents

We have experienced considerable lot-to-lot variation for both TF and tPA, as well as significant intra-lot variation between individual ampullas. In our experience, reconstituted TF and tPA are stable at −80 °C for a limited amount of time and should be stored for a maximum of six months. Thus, collected plasma samples within the same project should always be analyzed in batch using TF and tPA from the same reconstituted ampulla.

### 6.5. Preparation of Reagents Immediately Prior to Analysis

(A) Prepare a 60 µM solution of phospholipids in one step:Add 150 µL phospholipids 500 µM + 1100 µL HEPES buffer to the 5 mL tube marked “PL”. Add lid and vortex spin.(150 + 1100)/150 = 1:8.33 dilution. **500 µM/8.33 = 60 µM**.In well: 10 µL 60 µM phospholipids. (10 + 140)/10 = 1:15 dilution.**Final concentration in well: 60 µM/15 = 60 µM/15 = 4 µM**.

(B) Prepare a 1:665 dilution of TF in two steps:TF “A”: Add 10 µL TF + 1320 µL HEPES buffer to the 5 mL tube marked “TF A”. Add lid and vortex spin.(10 + 1320)/10 = 1:133 dilutionTF: Add 600 µL TF “A” + 2400 µL HEPES buffer to the separate 5 mL tube marked “TF”. Add lid and vortex spin.(600 + 2400)/600 = 1:5 dilution.**1:(133 × 5) = 1:655 dilution**.In well: 20 µL 1:665 TF. (20 + 130)/20 = 1:7.5 dilution.**Final dilution in well: 1:(665 × 7.5) = 1:4987 = 1:5000**.

(C) Prepare a 870 ng/mL tPA solution in two steps:tPA “A”: Add 40 µL tPA 100 µg/mL + 160 µL HEPES with 1% BSA to the 5 mL tube marked “tPA A”. Shake gently to mix.(40 + 160)/40 = 1:5 dilution.tPA: Add 150 µL tPA “A” + 3300 µL HEPES-BSA to the separate 5 mL tube marked “tPA”. Shake gently to mix.(150 + 3300)/150 = 1:23 dilution.**100 µg/mL/(5 × 23) = 0.879.6 µg/mL = 870 ng/mL**.In well: 20 µL 870 ng/mL tPA. (20 + 130)/20 = 1:7.5 dilution.**Final concentration in well: 870 ng/mL/7.5 = 115.9 ng/mL = 116 ng/mL**.

(D) Prepare a mix of phospholipids, TF and HEPES:Add 1200 µL HEPES, 1200 µL “PL” and 2400 µL “TF” to the 5 mL tube labelled “PLTFH”. Add lid and vortex spin.

### 6.6. Use of Software and Automatic Dispensers

Use of the WorkOut software and automatic dispensers greatly improves precision and timing and facilitates data analysis. If automatic dispensers are not an option at the laboratory, HEPES-Ca must be added manually to the plate; however, this inevitably leads to a delay between the activation of coagulation and reading. In this case, it is important to use a stopwatch during the final step of HEPES-Ca addition, noting the delay between HEPES-Ca addition and reading for each row. The time from HEPES-Ca addition to reading starts (after plate shaking is completed) should then be added to the final results. In our current software protocol, the delay from dispensation to reading starts is automatically added to the raw data and taken into account in the calculation of lag time.

If the WorkOut software is not an option for you, alternative free software can be used for generating a visual representation of curves and calculating derived parameters. e.g., the Shiny App Tool [37].

### 6.7. Setup of the Protocol in Workout

Slow kinetics. Absorbance 405 nm (0.1 s).Measure each plate 80 times. Delay between readings: 0 s.Plate: Flat bottomed. Generic and 12 size plate. Measure the height standard (min 8 mm).Temperature: 37 °C.Dispenser 1: 20 µL.Dispenser 2: 20 µL.Shaking: Slow, 10 s.Reading: Measurement mode: by plate.Baseline correction: mean from 0 to 500 s; consider to redefine from run to run.Crossing point (baseline corrected): 0.015 absorbance units.

### 6.8. Data Export

To create pictures: in the WorkOut protocol, click on well; click “Analysis”, right-click on picture “copy as image”. Open in MS Paint; save as .jpg.To export raw data:From Perkin Elmer 2030 Manager, choose “Explore protocols and results”.Find the folder where you stored the protocol and open folder.The results files are on the right side of this window. Open the wanted file.The file opens. After the Print button, you can see the Export button = two beams. Press this button. Save the file as .mht. Afterward, open in Excel and save as .xls.

## Figures and Tables

**Figure 1 mps-03-00067-f001:**
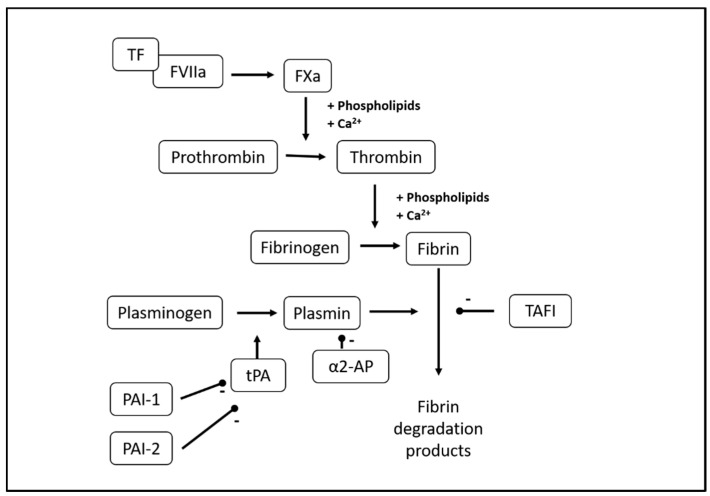
Overview of the fibrinolytic process. α2-AP, α2-antiplasmin; F, coagulation factor; PAI, plasminogen activator inhibitor; PL, phospholipids; TAFI, thrombin-activatable fibrinolysis inhibitor; TF, tissue factor; tPA, tissue plasminogen activator.

**Figure 2 mps-03-00067-f002:**
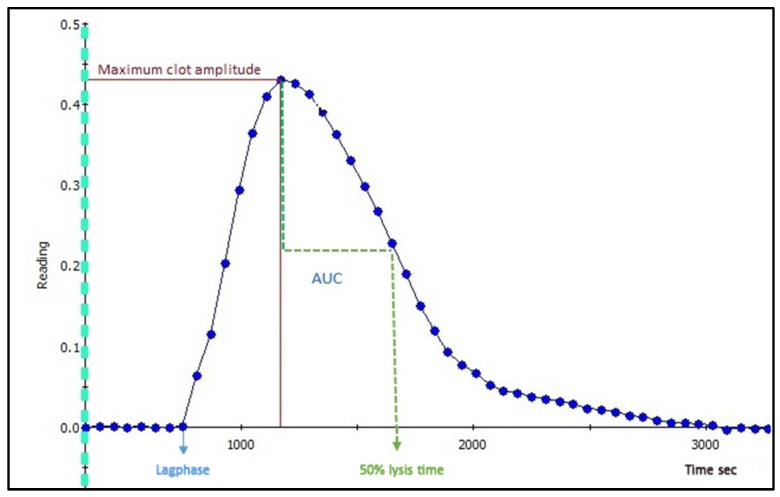
The clot-lysis curve and derived parameters.

**Figure 3 mps-03-00067-f003:**
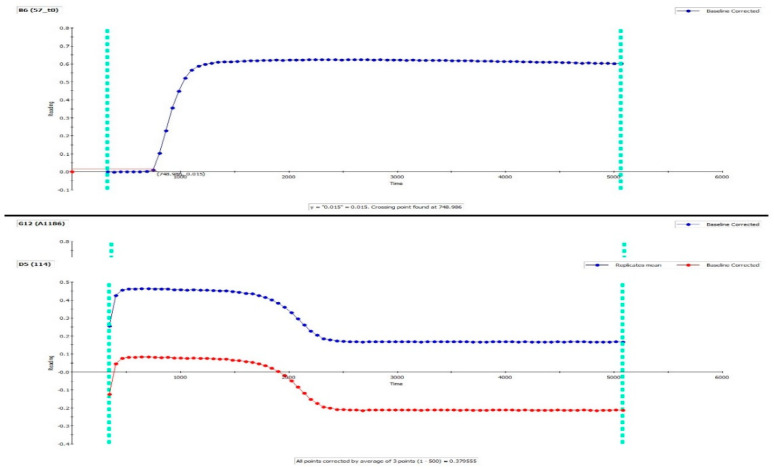
Top: Example of lysis resistance: only partial lysis is obtained. Bottom: Example of flat curve showing no net fibrin formation. Note the baseline correction.

**Table 1 mps-03-00067-t001:** Interpretation of clot-lysis results.

Parameter	Reference Interval [15]	Represents	Interpretation
Peak absorbance (AU)	0.18–0.74	Maximum fibrin concentration reached in well	↑: Increased procoagulant activity; high plasma fibrinogen ↓: Decreased procoagulant activity, may be due to low plasma levels of coagulation factors or fibrinogen
Integral (AU * s)	219–1051	Net fibrin formation Balance between fibrin formation and breakdown	↑: Increased procoagulant activity; decreased endogenous anticoagulant activity; decreased fibrinolytic capacity ↓: Decreased procoagulant activity or clot stability, may be due to low circulating coagulation factors, fibrinogen or factor XIII; increased fibrinolytic activity
50% lysis time (s)	309–1565	Time from maximum fibrin concentration is reached until 50% of the clot is lysed ^1^	↑: Decreased fibrinolytic capacity, may be due to low circulating plasminogen, high PAI-1 and/or TAFI [16] or anti-fibrinolytic treatment ↓: Increased fibrinolytic activity may be due to high plasma levels of tPA or uPA

Abbreviations: AU, absorbance units; PAI-1, plasminogen activator inhibitor 1; TAFI, thrombin-activatable fibrinolysis inhibitor; tPA/uPA; tissue/urokinase plasminogen activator. ^1^ Some authors calculate 50% lysis time as the time from the point where 50% of maximum fibrin formation is reached to the point where 50% of the clot is lysed [17].

**Table 2 mps-03-00067-t002:** Clot-lysis parameters with tissue factor vs. thrombin.

Parameter	Thrombin 0.03 U/mL	Tissue Factor 1:5000
Peak absorbance (AU)	0.32 (0.26–0.40)	0.68 (0.59–0.75)
Integral (AU × s)	408 (289–585)	1381 (1083–1733)
50% lysis time (s)	726 (570–912)	1483 (1154–1828)

N = 538. Median with interquartile range. Final concentration of tissue plasminogen activator (tPA) in well = 83 ng/mL.

**Table 3 mps-03-00067-t003:** Clot-lysis parameters with tissue plasminogen activator (tPA) 83 ng/mL vs. 116 ng/mL.

Parameter	tPA 83 ng/mL	tPA 116 ng/mL
Peak absorbance (AU)	0.69 (0.61–0.76)	0.68 (0.58–0.75)
Integral (AU × s)	1410 (1111–1748)	826 (665–1025)
50% lysis time (s)	1509 (1166–1830)	802 (653–1027)

N = 417. Median with interquartile range.

**Table 4 mps-03-00067-t004:** The clot-lysis assay in clinical conditions.

Condition	Findings
Cardiovascular disease	ACS: ↑ lysis time ACS patients vs. healthy controls [20]; ↑ lysis time at ACS associated with ↑ 1-year mortality [21] Stable CAD: ↑ lysis time in CAD patients with previous MI [22]; ↑ integral but not lysis time associated with subsequent poor cardiovascular outcome [23]
Ischaemic stroke	↑ lysis time in stroke patients at onset vs. healthy controls [24]; ↑ lysis time at onset associated with poor 3-month neurological function [25]
Venous thrombosis	↑ lysis time in DVT and PE patients compared with healthy controls [26,27]; in PE, ↑ lysis time associated with ↑ 12-month mortality [28] ↓ lysis time in patients with PE vs. patients with DVT alone [29] ↑ lysis time may predict VTE recurrence [30], though other studies found no association [16,31]
Diabetes mellitus	↑ integral and lysis time in CAD patients with type 2 diabetes vs. non-diabetic patients [32]
Hepatic dysfunction	Stable cirrhosis: uncertain; may vary according to etiology. ASH: ↓ lysis time; NASH: ↑ lysis time [33,34] ACLF: variable lysis times, influenced by concurrent factors [35] ALF: ↑↑ lysis time/lysis resistance
Sepsis	↑ lysis time in sepsis vs. healthy controls [36] and in septic vs. non-septic ACLF patients [35]; ↑ lysis time associated with lower platelet count but not survival [36] ↑ integral in sepsis vs. healthy controls; flat or lysis resistant clot-lysis curves associated with ↑ DIC and SOFA score *

Abbreviations: ACLF, cute-on-chronic liver failure; ACS, acute coronary syndrome; ALF, Acute liver failure; CAD, coronary artery disease; DIC, disseminated intravascular coagulation; DVT, deep vein thrombosis; MI, myocardial infarction; PE, pulmonary embolism SOFA, Sequential Organ Failure Assessment; VTE, venous thromboembolism * Larsen et al., unpublished data.

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
