# Peer review of "Fibrin Clot Formation and Lysis in Plasma"

_mps, 2020, doi:10.3390/mps3040067_

Round 1
Reviewer 1 Report
Reviewer comment:
- The manuscript entitled ,, Fibrin clot formation and lysis in plasma’’ provides interesting data on this topic. This is a nicely written contribution based on clearly described experiments. The topic of this study is interesting and in my opinion it could be interesting for a reasonable number of scientists.
It should be remembered that well-developed methods and protocols are the basis for the accuracy and repeatability of the obtained experimental results. When we care about accuracy and efficiency, having everything under control is fundamental. It makes sure that the results are as good as possible. Fully mastering the measurement techniques used will allow you to identify and avoid potential pitfalls that could ultimately prevent you from achieving your goals.
This work, thanks to marking critical points and comparing various methods, and on the basis of our own laboratory experience, definitely brings us closer to the perfection and repeatability of the described research method.
I have a editorial comments:
Bibliography:
- please complete the description of item 7 on lines 356-366:
- Saes JL, Schols SEM, van Heerde WL, Nijziel MR. Hemorrhagic disorders of fibrinolysis: a clinical review. J Thromb Haemost. 2018;16,1498-1509.
- please complete the description of item 35 on lines 437-439,
- Blasi A, Patel VC, Adelmeijer J, Azarian S, Hernandez Tejero M, Calvo A, et al. Mixed Fibrinolytic Phenotypes in Decompensated Cirrhosis and Acute-on-Chronic Liver Failure with Hypofibrinolysis in Those 438 With Complications and Poor Survival. Hepatology. 2020;71(4):1381-1390.
I suggest acceptance of this paper.
Reviewer 2 Report
The reviewed protocol is interesting, well, and clearly described.
The organization of a text is excellent.
I have only a few suggestions, mostly editorial nature.
Page 3 line 70
How long the sample can be stored under such conditions?
Page 3 line 83
Could you provide any information about the quality of water (for example its maximum permissible conductivity)
Page 7 line 201
Please remove “presented at the International Society of Thrombosis and Haemostasis Congress 2019” this information is not necessary and are included in the reference
